# Cream of the Crop: Distilling Prioritized Paths For One-Shot Neural Architecture Search

**Houwen Peng**[1*†]**, Hao Du**[2*]**, Hongyuan Yu**[3*]**, Qi Li**[4]**, Jing Liao**[2]**, Jianlong Fu**[1†]
[1]Microsoft Research Asia, [2]City University of Hong Kong,
[3]Chinese Academy of Sciences, [4]Tsinghua University

## Abstract

One-shot weight sharing methods have recently drawn great attention in neural architecture search due to high efficiency and competitive performance. However, weight sharing across models has an inherent deficiency, *i.e.*, insufficient training of subnetworks in hypernetworks. To alleviate this problem, we present a simple yet effective architecture distillation method. The central idea is that subnetworks can learn collaboratively and teach each other throughout the training process, aiming to boost the convergence of individual models. We introduce the concept of prioritized path, which refers to the architecture candidates exhibiting superior performance during training. Distilling knowledge from the prioritized paths is able to boost the training of subnetworks. Since the prioritized paths are changed on the fly depending on their performance and complexity, the final obtained paths are the cream of the crop. We directly select the most promising one from the prioritized paths as the final architecture, without using other complex search methods, such as reinforcement learning or evolution algorithms. The experiments on ImageNet verify such path distillation method can improve the convergence ratio and performance of the hypernetwork, as well as boosting the training of subnetworks. The discovered architectures achieve superior performance compared to the recent MobileNetV3 and EfficientNet families under aligned settings. Moreover, the experiments on object detection and more challenging search space show the generality and robustness of the proposed method. Code and models are available at https://github.com/microsoft/cream.git[2].

## 1 Introduction

Neural Architecture Search (NAS) is an exciting field which facilitates the automatic design of deep networks. It has achieved state-of-the-art performance on a variety of tasks, surpassing manually designed counterparts [*e.g.,* 1–3]. Recently, one-shot NAS methods became popular due to low computation overhead and competitive performance. Rather than training thousands of separate models from scratch, one-shot methods only train a single large hypernetwork capable of emulating any architecture in the search space. The weights are shared across architecture candidates, *i.e.*, subnetworks. Such strategy is able to reduce the search cost from thousands of GPU days to a few.

However, all architectures sharing a single set of weights cannot guarantee each individual subnetwork obtains sufficient training. Although one-shot models are typically only used to sort architectures in the search space, the capacity of weight sharing is still limited. As revealed by recent works [4], weight sharing degrades the ranking of architectures to the point of not reflecting their true performance, thus reducing the effectiveness of the search process. There are a few recent works addressing this issue from the perspective of knowledge distillation [5–7]. They commonly introduce a high-performing teacher network to boost the training of subnetworks. Nevertheless, these methods require the teacher model to be trained beforehand, such as a large pretrained model [5] or a third-party model [6]. This limits the flexibility of search algorithms, especially when the search tasks or data are entirely new and there may be no available teacher models.

---

[*]Equal contribution. Work done when Hao and Hongyuan were interns at MSRA. [†]Corresponding authors.
[2]We also provide another implementation based upon Microsoft NNI AutoML open source toolkit at here.

In this paper, we present *prioritized paths* to enable the knowledge transfer between architectures, without requiring an external teacher model. The core idea is that subnetworks can learn collaboratively and teach each other throughout the training process, and thus boosting the convergence of individual architectures. More specifically, we create *a prioritized path board* which recruits the subnetworks with superior performance as the internal teachers to facilitate the training of other models. The recruitment follows the selective competition principle, *i.e.*, selecting the superior and eliminating the inferior. Besides competition, there also exists collaboration. To enable the information transfer between architectures, we distill the knowledge from prioritized paths to subnetworks. Instead of learning from a fixed model, our method allows each subnetwork to select its best-matching prioritized path as the teacher based on the representation complementary. In particular, *a meta network* is introduced to mimic this path selection procedure. Throughout the course of subnetwork training, the meta network observes the subnetwork's performance on a held-out validation set, and learns to choose a prioritized path from the board so that if the subnetwork benefits from the prioritized path, the subnetwork will achieve better validation performance.

Such prioritized path distillation mechanism has three advantages. First, it does not require introducing third-party models, such as human-designed architectures, to serve as the teacher models, thus it is more flexible. Second, the matching between prioritized paths and subnetworks are meta-learned, which allows a subnetwork to select various prioritized paths to facilitates its learning. Last but not the least, after hypernetwork training, we can directly pick up the best performing architecture from the prioritized paths, instead of using either reinforcement learning or evolutionary algorithms to further search a final architecture from the large-scale hypernetwork.

The experiments demonstrate that our method achieves clear improvements over the strong baseline and establishes state-of-the-art performance on ImageNet. For instance, with the proposed prioritized path distillation, our search algorithm finds a 481M Flops model that achieving 79.2% top-1 accuracy on ImageNet. This model improves the SPOS baseline [8] by 4.5% while surpassing the EfficientNet-B0 [9] by 2.9%. Under the efficient computing settings, *i.e.*, Flops $\leq$ 100M, our models consistently outperform the MobileNetV3 [10], sometimes by nontrivial margins, *e.g.*, 3.0% under 43M Flops. The architecture discovered by our approach transfers well to downstream object detection task, getting an AP of 33.2 on COCO validation set, which is superior to the state-of-the-art MobileNetV3. In addition, distilling prioritized paths allows one-shot models to search architectures over more challenging search space, such as the combinations of MBConv [11], residual block [12] and normal 2D Conv, thus easing the restriction of designing a carefully constrained space.

## 2 Preliminary: One-Shot NAS

One-shot NAS approaches commonly adopt a weight sharing strategy to eschew training each subnetwork from scratch [13–15, 8, 16, among many others]. The architecture search space $\mathcal{A}$ is encoded in a hypernetwork, denoted as $\mathcal{N}(\mathcal{A}, W)$, where $W$ is the weight of the hypernetwork. The weight $W$ is shared across all the architecture candidates, *i.e.*, subnetworks $\alpha \in \mathcal{A}$ in $\mathcal{N}$. The search of the optimal architecture $\alpha^*$ in one-shot methods is formulated as a two-stage optimization problem. The first-stage is to optimize the weight $W$ by

$$W_{\mathcal{A}} = \arg\min_{W} \mathcal{L}_{train}(\mathcal{N}(\mathcal{A}, W)), \tag{1}$$

where $\mathcal{L}_{train}$ represents the loss function on training dataset. To reduce memory usage, one-shot methods usually sample subnetworks from $\mathcal{N}$ for optimization. We adopt the single-path uniform sampling strategy as the baseline, *i.e.*, each batch only sampling one random path from the hypernetwork for training [16, 8]. The second-stage is to search architectures via ranking the performance of subnetworks $\alpha \in \mathcal{A}$ based on the learned weights $W_{\mathcal{A}}$, which is formulated as

$$\alpha^* = \arg\max_{\alpha \in \mathcal{A}} Acc_{val}\left(\mathcal{N}(\alpha, w_{\alpha})\right), \tag{2}$$

where the sampled subnetwork $\alpha$ inherits the weight from $W_{\mathcal{A}}$ as $w_{\alpha}$, and $Acc_{val}^{\alpha}$ indicates the top-1 accuracy of the architecture $\alpha$ on validation dataset. Since that it is impossible to enumerate all the architectures $\alpha \in \mathcal{A}$ for evaluation, prior works resort to random search [16, 15], evolution algorithms [17, 8] or reinforcement learning [14, 18] to find the most promising one.

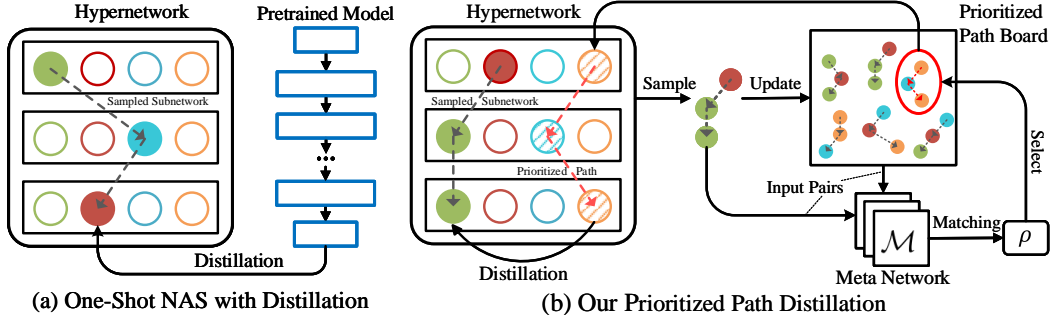

Figure 1: (a) Previous one-shot NAS methods use pretrained models for knowledge distillation. (b) Our prioritized path distillation enables knowledge transfer between architecture candidates. It contains three parts: hypernetwork, prioritized path board and meta network. The meta network is to select the best matching prioritized path to guide the training of the sampled subnetwork.

# 3 Distilling Priority Paths for One-Shot NAS

The weight sharing strategy reduces the search cost by orders of magnitude. However, it brings a potential issue, *i.e.*, the insufficient training of subnetworks within the hypernetwork. Such issue results in the performance of architectures ranked by the one-shot weight is weakly correlated with the true performance. Thus, the search based on the weight $W$ may not find a promising architecture. To boost the training of subnetworks, we present *prioritized path distillation*. The intuitive idea is to leverage the well-performing subnetwork to teach the under-trained ones, such that all architectures converge to better solutions. In the following, we first present the mechanism of prioritized path board, which plays a fundamental role in our approach. Then, we describe the search algorithm using the prioritized paths and knowledge transfer between architectures. The overall framework is visualized in Fig. 1.

## 3.1 Prioritized Path Board

Prioritized paths refer to the architecture candidates which exhibit promising performance during hypernetwork training. The prioritized path board is an architecture set which contains $K$ prioritized paths $\hat{\alpha}_k$, *i.e.*, $\mathbb{B}=\{\hat{\alpha}_k\}_{k=1}^{K}$. The board is first initialized with random paths, and then changed on the fly depending on the path performance. More specifically, for each batch, we randomly sample a single path $\alpha$ from the hypernetwork $\mathcal{N}$ and train the path to update the corresponding shared-weight $w_\alpha$. After that, we evaluate the path $\alpha$ on the validation dataset (a subset is used to save computation cost), and get its performance $Acc_{val}(\mathcal{N}(\alpha, w_\alpha))$. If the current path $\alpha$ performs superior than the least competitive prioritized paths in $\mathbb{B}$, then it will replace that prioritized path as

$$\hat{\alpha}_k \leftarrow \{\alpha \mid Acc_{val}((\mathcal{N}(\alpha, w_\alpha)) \geq Acc_{val}(\mathcal{N}(\hat{\alpha}_k, w_{\hat{\alpha}_k})) \And Flops(\alpha) \leq Flops(\hat{\alpha}_k), \ \hat{\alpha}_k \in \mathbb{B}\}, \quad (3)$$

where $Flops(\cdot)$ counts the multiply-add operations in models. Eq. (3) indicates the update of prioritized path board follows the selective competition, *i.e.*, selecting models with higher performance and lower complexity. Thus, the prioritized paths are changed on the fly. The final left paths on the board $\mathbb{B}$ are the Pareto optima [19] among all the sampled paths during the training process.

## 3.2 Architecture Search with Prioritized Paths

Our solution to the insufficient training of subnetworks is to distill the knowledge from prioritized paths to the weight sharing subnetworks. Due to the large scale of the search space, the structure of subnetworks are extremely diverse. Some subnetworks may be beneficial to other peer architectures, while others may not or even harmful. Hence, we allow each subnetwork to find its best matching collaborator from the prioritized path board, such that the matched path can make up its deficiency. We propose to learn the matching between prioritized paths and subnetworks by a meta network $\mathcal{M}$. Since there is no available groundtruth to measure the matching degree of two architectures, we use the learning state (*i.e.*, validation loss) of subnetworks as the signal to supervise the learning of the meta network. The underlying reason is that if the gradient updates of the meta network encourage the subnetworks to learn from the selected prioritized path and achieve a small validate loss, then this matching is profitable.

The hypernetwork training with prioritized path distillation includes three iterative phases.

---

**Algorithm 1** Architecture Search with Prioritized Paths

---

**Input:** Training and validation data, hypernetwork $\mathcal{N}$ with weight $W$, meta network $\mathcal{M}$ with weight $\theta$ and its update interval $\tau$, path board $\mathbb{B}$, max iteration $T$, user specified *min* and *max* Flops.
**Output:** The most promising architecture.
1: Random initialize $W$, $\theta$, $\mathbb{B}$ with path $Flops \in [min, max]$
2: **while** search step $t \in [0, T]$ and not converged **do**
3:     Randomly sample a path $\alpha$ from $\mathcal{N}$
4:     Select the best fit path $\hat{\alpha}_k^*$ in $\mathbb{B}$ according to Eq. (4)
5:     Calculate the loss $\mathcal{L}_{CE}$ and $\mathcal{L}_{KD}$ over one *train* batch
6:     Update the weight $w_\alpha^{(t)}$ of path $\alpha$ according to Eq. (5)
7:     Calculate $Flops(\alpha)$ and top-1 accuracy $Acc_\alpha$ on *val* subset
8:     Update $\mathbb{B}$ according to Eq. (3)
9:     **if** $t \ Mod \ \tau = 0$ **then**
10:         Calculate loss $\mathcal{L}_{CE}$ on *val* dataset with the updated weight $w_\alpha^{(t+1)}$ according to Eq. (6)
11:         Update the weight $\theta$ of meta network $\mathcal{M}$ by calculating $\nabla_\theta \mathcal{R}$
12:     **end if**
13: **end while**
14: Select the best performing architecture from $\mathbb{B}$ on validation dataset.

---

**Phase 1: choosing the prioritized path.** For each batch, we first randomly sample a subnetwork $\alpha$. Then, we use the meta network $\mathcal{M}$ to select the best fit model from the prioritized path board $\mathbb{B}$, aiming to facilitate the training of the sampled path. The selection is formulated as

$$\hat{\alpha}_k^* = \underset{\hat{\alpha}_k \in \mathbb{B}}{\arg\max} \ \{\rho \mid \rho = \mathcal{M}( \ (\mathcal{N}(x, \hat{\alpha}_k, w_{\hat{\alpha}_t}) - \mathcal{N}(x, \alpha, w_\alpha)), \ \theta \ ) \}, \tag{4}$$

where $\rho$ is the output of the meta network $\mathcal{M}$ and represents the matching degree (the higher the better) between the prioritized path $\hat{\alpha}_k$ and the subnetwork $\alpha$, $x$ indicates the training data, and $\theta$ denotes the weight of $\mathcal{M}$. The input to the meta network $\mathcal{M}$ is the difference of the feature logits between the subnetworks $\mathcal{N}(\hat{\alpha}_k, w_{\hat{\alpha}_k})$ and $\mathcal{N}(\alpha, w_\alpha)$. Such difference reflects the complementarity of the two paths. The meta network learns to select the prioritize path $\hat{\alpha}_k^*$ that is complementary to the current subnetwork $\alpha$.

**Phase 2: distilling knowledge from the prioritized path.** With the picked prioritized path $\hat{\alpha}_k^*$, we perform knowledge distillation to boost the training of the subnetwork $\alpha$. The distillation is supervised by a weighted average of two different objective functions. The first objective function $\mathcal{L}_{CE}$ is the cross entropy with the correct labels $y$. This is computed using the logits in softmax of the subnetwork, *i.e.*, $p(x, w_\alpha) = softmax(\mathcal{N}(x, \alpha, w_\alpha))$. The second objective function $\mathcal{L}_{KD}$ is the cross entropy with the soft target labels and this cross entropy distills knowledge from the prioritized path $\hat{\alpha}_k^*$ to the subnetwork $\alpha$. The soft targets $q(x, w_{\hat{\alpha}_k^*})$ are generated by a softmax function that converts feature logits to a probability distribution. We use SGD with a learning rate $\eta$ to optimize the objective functions and update the subnetwork weight $w_\alpha$ as

$$w_\alpha^{(t+1)} = w_\alpha^{(t)} - \eta \nabla_{w_\alpha}( \ \mathcal{L}_{CE}(y, p(x, w_\alpha^{(t)})) + \rho \mathcal{L}_{KD}(q(x, w_{\hat{\alpha}_k^*}), p(x, w_\alpha^{(t)})) \ )|_{w_\alpha^{(t)}}, \tag{5}$$

where $t$ is the iteration index. It is worth noting that we use the matching degree $\rho$ as the weight for the distillation objective function. The underlying reason is if the selected prioritized path are well-matched to the current path, then it can play an more important role to facilitate the learning, and vise versa. After the weight update, we evaluate the performance of the subnetwork $\alpha$ on the validation subset and calculate its model complexity. If both performance and complexity satisfy Eq. (3), then the path $\alpha$ is added into the prioritized path board $\mathbb{B}$.

**Phase 3: updating the meta network.** Since there is no available groundtruth label measuring the matching degree and complementarity of two architectures, we resort to the loss of the subnetwork to guide the training of the matching network $\mathcal{M}$. The underlying reason is that if one prioritized path $\hat{\alpha}_k^*$ is complementary to the current subnetwork $\alpha$, then the updated subnetwork with the weight $w_\alpha^{(t+1)}$ can achieve a lower loss on the validation data. We evaluate the new weight $w_\alpha^{(t+1)}$ on the validatation data $(x, y)$ using the cross entropy loss $\mathcal{L}_{CE}(y, p(x, w_\alpha^{(t+1)}))$. Since $w_\alpha^{(t+1)}$ depends on $\rho$ via Eq. (5) while $\rho$ depends on $\theta$ via Eq. (4), this validation cross entropy loss is a function of $\theta$. Specifically, dropping $(x, y)$ from the equations for readability, we can write:

$$\begin{aligned} \mathcal{L}_{CE}(y, p(x, w_\alpha^{(t+1)})) &\triangleq \mathcal{R}(w_\alpha^{(t+1)}) \\ &= \mathcal{R}(w_\alpha^{(t)} - \eta \nabla_{w_\alpha}( \ \mathcal{L}_{CE}(y, p(x, w_\alpha^{(t)})) + \boxed{\rho} \mathcal{L}_{KD}(q(x, w_{\hat{\alpha}_k^*}), p(x, w_\alpha^{(t)})) \ )|_{w_\alpha^{(t)}}). \end{aligned} \tag{6}$$

This dependency allows us to compute $\nabla_\theta \mathcal{R}$ to update $\theta$ and minimize $\mathcal{R}(w_\alpha^{(t+1)})$. The differentiation $\nabla_\theta \mathcal{R}$ requires computing the gradient of gradient, which is time-consuming, we thereby updates $\theta$ every $\tau$ iterations. In essence, the meta network observing the subnetwork's validation loss to improve itself is similar to an agent in reinforcement learning performing on-policy sampling and learning from its own rewards [20]. In implementation, we adopts one fully-connected layer with 1,000 hidden nodes as the architecture of meta network, which is simple and efficient.

The above three phases are performed iteratively to train the hypernetwork. The iterative procedure is outlined in Alg. 1. Thanks to the prioritized path distillation mechanism, after hypernetwork training, we can directly select the best performing subnetwork from the prioritized path board as the final architecture, instead of further performing search on the hypernetwork.

## 4 Experiments

In this section, we first present ablation studies dissecting our method on image classification task, and then compare our method with state-of-the-art NAS algorithms. The experiments on object detection and more challenging search space are performed to evaluate the generality and robustness.

### 4.1 Implementation Details

**Search space**. Similar to recent works [9, 10, 5–7], we perform architecture search over the search space consisting of mobile inverted bottleneck MBConv [11] and squeeze-excitation modules [21] for fair comparisons. There are seven basic operators, including MBConv with kernel sizes of {3,5,7} and expansion rates of {4,6}, and an additional skip connection to enable elastic depth of architectures. The space contains about $3.69 \times 10^{16}$ architecture candidates in total.
**Hypernetwork**. Our hypernetwork is similar to the baseline SPOS [8]. The architecture details are presented in Appendix A of the *supplementary materials*. We train the hypernetwork for 120 epochs using the following settings: SGD optimizer [22] with momentum 0.9 and weight decay 4e-5, initial learning rate 0.5 with a linear annealing. The meta network is updated every $\tau$=200 iterations to save computation cost. The number of prioritized paths $K$ is empirically set to 10, while the number of images sampled from validation set for prioritized path selection in Eq. (3) is set to 2,048.
**Retrain**. We retrain the discovered architectures for 500 epochs on Imagenet using similar settings as EfficientNet [9]: RMSProp optimizer with momentum 0.9 and decay 0.9, weight decay 1e-5, dropout ratio 0.2, initial learning rate 0.064 with a warmup [23] in the first 3 epochs and a cosine annealing, AutoAugment [24] policy and exponential moving average are adopted for training. We use 16 Nvidia Tesla V100 GPUs with a batch size of 2,048 for the retrain.

### 4.2 Ablation Study

We dissect our method and evaluate the effects of each components. Our baseline is the single-path one-shot method, which trains the hypernetwork with uniform sampling and searches architectures by an evolution algorithm [8]. We re-implement this algorithm in our codebase, and it achieves 76.3% top-1 accuracy on ImageNet, being superior to the original 74.7% reported in [8] due to different search spaces (ShuffleUnits [8] *v.s.* MBConv [11]). If we replace the evolution search with the proposed prioritized path mechanism, the performance is still comparable to the baseline, as presented in Tab. 1(#1 *v.s.* #2). This suggests the effectiveness of the prioritized paths. By comparing #2 with #4/#5, we observe that the knowledge distillation between prioritized paths and subnetworks is indeed helpful for both hypernetwork training and the final performance, even when the matching between prioritized paths and subnetworks is random, *i.e.* #4. The meta-learned matching function is superior to random matching by 1.3% in terms of top-1 accuracy on ImageNet. The ablation between #5 and #6 shows that the evolution search over the hypernetwork performs comparably to

| # | Single-path Training | Evolution Alg. | Priority Path | Fixed Match | Random Match | Meta Match | Kendall Rank on subImageNet | Hypernet on ImageNet | Top-1 Acc on ImageNet | Model FLOPS |
|---|---|---|---|---|---|---|---|---|---|---|
| 1 | ✓ | ✓ | | | | | 0.19 | 63.5 | 76.3 | 450M |
| 2 | ✓ | | ✓ | | | | 0.19 | 63.5 | 76.5 | 433M |
| 3 | ✓ | | | ✓ | | | 0.23 | 64.9 | 77.6 | 432M |
| 4 | ✓ | | ✓ | | ✓ | | 0.25 | 65.4 | 77.9 | 451M |
| 5 | ✓ | | ✓ | | | ✓ | 0.37 | 67.0 | 79.2 | 481M |
| 6 | ✓ | ✓ | ✓ | | | ✓ | 0.37 | 67.0 | 79.2 | 487M |

Table 1: Component-wise analysis. *Fixed, Random* and *Meta* matching represent performing distillation with the largest subnetwork, random sampled prioritized path and meta-learned prioritized path, respectively.

Table 2: Ablation for the number of prioritized paths.

| Board Size $K$ | 1 | 5 | 10 | 20 | 50 |
|---|---|---|---|---|---|
| Hypernetwork (Top-1) | 65.4 | 65.9 | 67.0 | 67.3 | 67.5 |
| Search Cost (GPU days) | 9 | 10 | 12 | 16 | 27 |

Table 3: Ablation for the number of *val* images.

| Image Numbers | 0.5k | 1k | 2k | 5k | 10k | 50k |
|---|---|---|---|---|---|---|
| Kendall Rank (Top-1) | 0.72 | 0.74 | 0.75 | 0.85 | 0.94 | 1 |
| Kendall Rank (Top-5) | 0.47 | 0.50 | 0.66 | 0.76 | 0.89 | 1 |

Figure 2: Comparison with state-of-the-art methods on ImageNet under mobile settings (Flops≤600M).

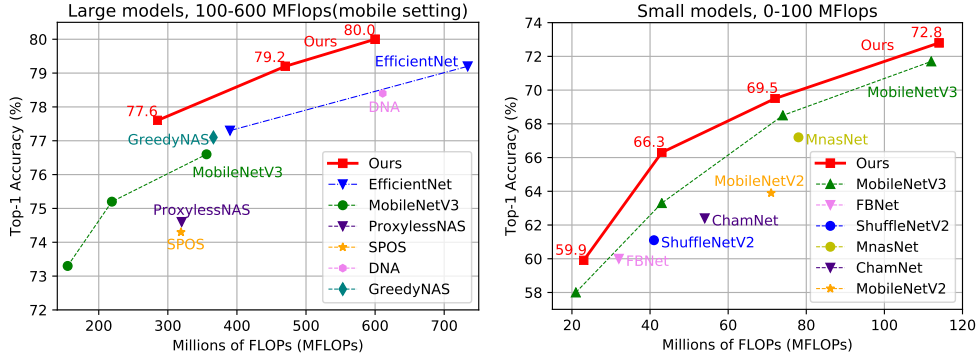

the prioritized path distillation, suggesting that the final paths left in the prioritized path board is the "cream of the crop".

We further perform a correlation analysis to evaluate whether the enhanced training of the hypernetwork can improve the ranking of subnetworks. To this end, we randomly sample 30 subnetworks and calculate the rank correlation between the weight sharing performance and the true performance of training from scratch. Unfortunately, training such many subnetworks on ImageNet is very computational expensive, we thus construct a subImageNet dataset, which only consists of 100 classes randomly sampled from ImageNet. Each class has 250 training images and 50 validation images (Image lists are released with the code). Its size is about 50× smaller than the original ImageNet. The Kendall rank correlation coefficient [25] on subImageNet is reported in Tab. 1. It is clear to observe that after performing prioritized path distillation, the ranking correlation is improved significantly, *e.g.*, from the baseline 0.19 to 0.37 (#1 *v.s.* #5 in Tab. 1).

There are two hyperparameters in our method: one is the size of the prioritized path board and the other is the number of validation images for prioritized path selection in Eq. (3). The impact of these two hyperparameters are reported in Tab. 2 and 3 respectively. We observe that when the number of prioritized paths is increased, the performance of hypernetwork becomes better, yet bringing more search overhead. Considering the tradeoff, we empirically set the number of prioritized paths to 10 in the experiments. A similar situation is occurred on the number of *val* images. We randomly sample 2,048 images from the validation set (50k images in total) for prioritized path selection because it allows fast evaluations and keep a relatively high Kendall rank.

### 4.3   Comparison with State-of-the-Art NAS Methods

Fig. 2 presents the comparison of our method with state-of-the-arts under mobile settings on ImageNet. It shows that when the model Flops are smaller than 600M, our method consistently outperforms the recent MobileNetV3 [10] and EfficientNet-B0/B1 [9]. In particular, our method achieves 77.6% top-1 accuracy on ImageNet with 287M Flops, which is 1.0% higher than MobileNetV3$_{1.25x}$ while using 1.2× fewer Flops. Moreover, our method is flexible to search low complexity models, only requiring the users input a desired minimum and maximum Flops constraint. From Fig. 2(right), we can see that when the Flops are smaller than 100M, our models establishes new state-of-the-arts. For example, when using 43M Flops, MobileNetV3 is inferior to our model by 3.0%. Besides model complexity, we are also interested in inference latency. As shown in Tab. 4, where we report the average latency of 1,000 runs, our method runs 1.1× faster than EfficientNet-B0 [9] and 1.7× faster than MobileNetV3 on a single core of Intel Xeon CPU E5-2690. Also, the performance of our

| | Acc. @ Latency | | | Acc. @ Latency |
|---|---|---|---|---|
| EfficientNet-B0 [9] | 76.3% @ 96ms | | MobileNetV3$_{small}$ [10] | 51.7% @ 15ms |
| Ours (287M Flops) | 77.6% @ 89ms | | Ours (14M Flops) | 53.8% @ 9ms |
| **Speedup** | **1.1x** | | **Speedup** | **1.7x** |

Table 4: Inference latency comparison. Latency is measured with batch size 1 on a single core of Intel Xeon CPU E5-2690.

Table 5: Comparison of state-of-the-art NAS methods on ImageNet. †: TPU days, ⋆: reported by [8], ‡: searched on CIFAR-10.

| | Methods | Top-1 (%) | Top-5 (%) | Flops (M) | Params (M) | Memory cost | Hypernet train (GPU days) | Search cost (GPU days) |
|---|---|---|---|---|---|---|---|---|
| 200 – 350M Flops | MobileNetV3$_{Large1.0}$ [10] | 75.2 | - | 219 | 5.3 | single path | 288† | - |
| | OFA [5] | 76.9 | - | 230 | - | two paths | 53 | 2 |
| | AKD [26] | 73.0 | 92.2 | 300 | - | single path | - | 1000† |
| | MobileNetV2 [11] | 72.0 | 91.0 | 300 | 3.4 | - | - | - |
| | MnasNet-A1 [18] | 75.2 | 92.5 | 312 | 3.9 | single path | 288† | - |
| | FairNAS-C [27] | 74.7 | 92.1 | 321 | 4.4 | single path | 10 | 2 |
| | SPOS [8] | 74.7 | - | 328 | - | single path | 12 | < 1 |
| | **Cream-S (Ours)** | **77.6** | **93.3** | 287 | 6.0 | two paths | 12 | 0.02 |
| 350 – 500M | SCARLET-A [27] | 76.9 | 93.4 | 365 | 6.7 | single path | 10 | 12 |
| | GreedyNAS-A [28] | 77.1 | 93.3 | 366 | 6.5 | single path | 7 | < 1 |
| | EfficientNet-B0 [9] | 76.3 | 93.2 | 390 | 5.3 | - | - | - |
| | ProxylessNAS [29] | 75.1 | - | 465 | 7.1 | two paths | 15⋆ | - |
| | **Cream-M (Ours)** | **79.2** | **94.2** | 481 | 7.7 | two paths | 12 | 0.02 |
| 500 – 600M | DARTS [30] | 73.3 | 91.3 | 574 | 4.7 | whole hypernet | 4‡ | - |
| | BigNASModel-L [7] | 79.5 | - | 586 | 6.4 | two paths | 96† | - |
| | OFA$_{Large}$ [5] | 80.0 | - | 595 | - | two paths | 53 | 2 |
| | DNA-d [6] | 78.4 | 94.0 | 611 | 6.4 | single path | 24 | 0.6 |
| | EfficientNet-B1 [9] | 79.2 | 94.5 | 734 | 7.8 | - | - | - |
| | **Cream-L (Ours)** | **80.0** | **94.7** | 604 | 9.7 | two paths | 12 | 0.02 |

method is 1.3% superior to EfficientNett-B0 and 2.4% superior to MobileNetV3. This suggests our models are competitive when deployed on real hardwares.

Tab. 5 presents more comparisons. It is worth noting that there are few recent works leveraging knowledge distillation techniques to boost training [5–7]. Compared to these methods, our prioritized path distillation is also superior. Specifically, DNA [6] recruits EfficientNet-B7 [6], a very high-performance third-party model, as the teacher and achieves 78.4% top-1 accuracy (without using AutoAugment), while our method (Cream-L) gets a superior accuracy of 80.0% without using any other pretrained models. Our method performs comparably to the recent OFA [5] yet taking much less time on hypernetwork training, *i.e.*, 12 *v.s.* 53 GPU days. Thanks to the prioritized path mechanism, our method only need to evaluate $K$=10 prioritized paths on the validation set and then select the best performing one. This procedure only takes 0.02 GPU days, which is 30× faster than other approaches of using evolutionary search algorithm, such as SPOS [8] and OFA [5]. The learned architectures are plotted in *Appendix B*.

### 4.4 Generality and Robustness

To further evaluate the generalizability of the architecture found by our method, we transfer it to the downstream object detection task. We use the discovered architecture as a drop-in replace-ment for the backbone feature extractor in RetinaNet [31] and compare it with other backbone networks on COCO dataset [32]. We perform the training on `train2017` set (~118k images) and the evaluation on `val2017` set (5k images) with 32 batch size on 8 V100 GPUs. The same as [27], we train the detection model by 12 epochs. The initial learning rate is 0.04 and multiplied by 0.1 at the epochs 8 and 11. The optimizer is SGD with 0.9 momentum and 1e-4 weight decay.

As shown in Tab. 7, our method surpasses Mo-bileNetV2 by 4.9% while using fewer Flops. Compared to MnasNet [18], our method utilizes 19% fewer Flops while achieving 2.7% higher performance, suggesting the architecture has good generalization capacity when transferred to other vision tasks. If we further increase the model complexity, our method can achieve an AP of 36.8%, which is comparable to the recent Hit-Detector [33] (36.9AP with 1839M Flops) but uses much less Flops.

| Search Space | Method | Kendall Rank† | Top-1‡ |
|---|---|---|---|
| MBconv [10] | SPOS[8] | 0.19 | 75.8 |
| | Ours | 0.37 | 77.7 |
| +ResBlock [12] | SPOS[8] | 0.09 | 75.0 |
| | Ours | 0.28 | 77.2 |
| + 2D Conv | SPOS[8] | 0.04 | 74.0 |
| | Ours | 0.25 | 77.1 |

Table 6: Search on different space. †: calculated on subImageNet using the sampled 30 subnetworks. ‡: re-trained on ImageNet with *120 epochs*.

A robust search algorithm should be capable of searching architectures over diverse search spaces. To evaluate this, we evaluate our method on more challenging space, *i.e.*, the combinations of operators from different designed space, including

Table 7: Object detection results of various drop-in backbones on the COCO *val2017*. Top-1 represents the top-1 accuracy on ImageNet. †: reported by [27].

| Backbones | FLOPs (M) | AP (%) | $AP_{50}$ | $AP_{75}$ | $AP_S$ | $AP_M$ | $AP_L$ | Top-1 (%) |
|---|---|---|---|---|---|---|---|---|
| MobileNetV2† [11] | 300 | 28.3 | 46.7 | 29.3 | 14.8 | 30.7 | 38.1 | 72.0 |
| SPOS† [8] | 365 | 30.7 | 49.8 | 32.2 | 15.4 | 33.9 | 41.6 | 75.0 |
| MnasNet-A2† [18] | 340 | 30.5 | 50.2 | 32.0 | 16.6 | 34.1 | 41.1 | 75.6 |
| MobileNetV3† [10] | 219 | 29.9 | 49.3 | 30.8 | 14.9 | 33.3 | 41.1 | 75.2 |
| MixNet-M† [34] | 360 | 31.3 | 51.7 | 32.4 | 17.0 | 35.0 | 41.9 | 77.0 |
| FairNAS-C [27] | 325 | 31.2 | 50.8 | 32.7 | 16.3 | 34.4 | 42.3 | 76.7 |
| MixPath-A [35] | 349 | 31.5 | 51.3 | 33.2 | 17.4 | 35.3 | 41.8 | 76.9 |
| **Cream-S (Ours)** | 287 | **33.2** | **53.6** | **34.9** | **18.2** | **36.6** | **44.4** | **77.6** |

MBConv [10], Residual Block [12] and normal 2D convolutions. Due to limited space, we present the detailed settings of the new search spaces in Appendix C. As the results reported in Tab. 6, we observe that when the search space becomes more challenging, the performance of the baseline SPOS algorithm [8] is degraded. In contrast, our method shows relatively stable performance, demonstrating it has potentials to search for architectures over more flexible spaces. The main reason is attributed to the prioritised path distillation, which improves the ranking correlation of architectures.

## 5  Related Work

**Neural Architecture Search**. Early NAS approaches search a network using either reinforcement learning [1, 36] or evolution algorithms [37, 17]. These approaches require training thousands of architecture candidates from scratch, leading to unaffordable computation overhead. Most recent works resort to the one-shot weight sharing strategy to amortize the searching cost [16, 14, 13, 8]. The key idea is to train a single over-parameterized hypernetwork model, and then share the weights across subnetworks. The training of hypernetwork commonly samples subnetwork paths for optimization. There are several path sampling methods, such as drop path [15], single path [8, 16] and multiple paths [28, 27]. Among them, single-path one-shot model is simple and representative. In each iteration, it only samples one random path and train the path using one batch data. Once the training process is finished, the subnetworks can be ranked by the shared weights. On the other hand, instead of searching over a discrete set of architecture candidates, differentiable methods [30, 38, 29] relax the search space to be continuous, such that the search can be optimized by the efficient gradient descent. Recent surveys on architecture search can be found in [39, 40].

**Distillation between Architectures**. Knowledge distillation [41] is a widely used technique for information transfer. It compresses the "dark knowledge" of a well trained larger model to a smaller one. Recently, in one-shot NAS, there are few works leveraging this technique to boost the training of hypernetwork [*e.g.,* 42], and they commonly introduce additional large models as teachers. More specifically, OFA [5] pretrains the largest model in the search space and use it to guide the training of other subnetworks, while DNA [6] employs the third-party EfficientNet-B7 [9] as the teacher model. These search algorithms will become infeasible if there is no available pretrained model, especially when the search task and data are entirely new. The most recent work, *i.e.* BigNAS [7], proposes inplace distillation with a sandwich rule to supervise the training of subnetworks by the largest child model. Although this method does not reply on other pretrained models, it cannot guarantee the fixed largest model is the best teacher for all other subnetworks. Sometimes the largest model may be a noise in the search space. In contrast, our method dynamically recruits prioritized paths from the search space as the teachers, and it allows subnetworks to select their best matching prioritized models for knowledge distillation. Moreover, after training, the prioritized paths in our method can serve as the final architectures directly, without requiring further search on the hypernetwork.

## 6  Conclusions

In this work, motivated by the insufficient training of subnetworks in the weight sharing methods, we propose prioritized path distillation to enable knowledge transfer between architectures. Extensive experiments demonstrate the proposed search algorithm can improve the training of the weight sharing hypernetwork and find promising architectures. In future work, we will consider adding more constraints on prioritized path selection, such as both model size and latency, thus improving the flexibility and user-friendliness of the search method. The theoretical analysis of the prioritized path distillation for weight sharing training is another potential research direction.

# 7 Broader Impact

Similar to previous NAS works, this work does not have immediate societal impact, since the algorithm is only designed for image classification, but it can indirectly impact society. As an example, our work may inspire the creation of new algorithms and applications with direct societal implications. Moreover, compared with other NAS methods that require additional teacher model to guide the training process, our method does not need any external teacher models. So our method can be used in a closed data system, ensuring the privacy of user data.

# 8 Acknowledgements

We acknowledge the anonymous reviewers for their insightful suggestions. In particular, we would like to thank Microsoft OpenPAI v-team for providing AI computing platform and large-scale jobs scheduling support, and Microsoft NNI v-team for AutoML toolkit support as well as helpful discussions and collaborations. Jing Liao and Hao Du were supported in part by the Hong Kong Research Grants Council (RGC) Early Career Scheme under Grant 9048148 (CityU 21209119), and in part by the CityU of Hong Kong under APRC Grant 9610488. This work was led by Houwen Peng, who is the Primary Contact (✉ houwen.peng@microsoft.com).

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
