[Supplementary Material]

# Cream of the Crop: Distilling Prioritized Paths For One-Shot Neural Architecture Search
## ——— Supplementary Material ———

**Appendix A**

| Input Shape | Operators | Channels | Repeat | Stride |
|---|---|---|---|---|
| $224^2 \times 3$ | $3 \times 3$ Conv | 16 | 1 | 2 |
| $112^2 \times 16$ | $3 \times 3$ Depthwise Separable Conv | 16 | 1 | 2 |
| $56^2 \times 16$ | MBConv / SkipConnect | 24 | 4-6 | 2 |
| $28^2 \times 24$ | MBConv / SkipConnect | 40 | 4-6 | 2 |
| $14^2 \times 40$ | MBConv / SkipConnect | 80 | 4-6 | 1 |
| $14^2 \times 80$ | MBConv / SkipConnect | 96 | 4-6 | 2 |
| $7^2 \times 96$ | MBConv / SkipConnect | 192 | 4-6 | 1 |
| $7^2 \times 192$ | $1 \times 1$ Conv | 320 | 1 | 1 |
| $7^2 \times 320$ | Global Avg. Pooling | 320 | 1 | 1 |
| 320 | $1 \times 1$ Conv | 1,280 | 1 | 1 |
| $1,280$ | Fully Connect | 1,000 | 1 | - |

Table 1: The structure of the hypernetwork. The "MBConv" contains 6 inverted bottleneck residual block MBConv [1] ( kernel sizes of {3,5,7}) with the squeeze and excitation module (expansion rates {4,6}). The "Repeat" represents the maximum number of repeated blocks in a group. The "Stride" indicates the convolutional stride of the first block in each repeated group. When searching for tiny models, the Input Shape will be decreased to reduce Flops.

# Appendix B

**(a) Ours(114M)**

3x224x224 → Stem → 16x112x112 → DS 4 3x3 SE → 16x112x112 → MB 6 5x5 SE → 24x56x56 → MB 6 5x5 SE → 40x28x28 → MB 6 5x5 SE → 40x28x28 → MB 6 5x5 SE → 80x14x14 → MB 6 5x5 SE → 80x14x14 → MB 6 5x5 SE → 96x14x14 → MB 6 5x5 SE → 96x14x14 → MB 6 5x5 SE → 96x14x14 → MB 6 5x5 SE → 192x7x7 → MB 6 5x5 SE → 192x7x7 → Conv 1x1 → 320x7x7 → Pooling+FC

**(c) Ours-S(287M)**

3x224x224 → Stem → 16x112x112 → DS 4 3x3 SE → 16x112x112 → MB 6 5x5 SE → 24x56x56 → MB 6 5x5 SE → 40x28x28 → MB 6 5x5 SE → 40x28x28 → MB 6 5x5 SE → 80x14x14 → MB 6 3x3 SE → 80x14x14 → MB 6 5x5 SE → 80x14x14 → MB 6 5x5 SE → 96x14x14 → MB 6 5x5 SE → 96x14x14 → MB 6 5x5 SE → 96x14x14 → MB 6 5x5 SE → 192x7x7 → MB 6 5x5 SE → 192x7x7 → MB 6 5x5 SE → 192x7x7 → Conv 1x1 → 320x7x7 → Pooling+FC

**(f) Ours-M(481M)**

3x224x224 → Stem → 16x112x112 → DS 4 3x3 SE → 16x112x112 → MB 6 5x5 SE → 24x56x56 → MB 4 7x7 SE → 24x56x56 → MB 6 5x5 SE → 24x56x56 → MB 6 3x3 SE → 24x56x56 → MB 6 5x5 SE → 40x28x28 → MB 4 5x5 SE → 40x28x28 → MB 6 5x5 SE → 40x28x28 → MB 4 3x3 SE → 40x28x28 → MB 6 5x5 SE → 80x14x14 → MB 6 5x5 SE → 80x14x14 → MB 6 5x5 SE → 80x14x14 → MB 6 3x3 SE → 80x14x14 → MB 6 3x3 SE → 80x14x14 → MB 6 5x5 SE → 96x14x14 → MB 6 5x5 SE → 96x14x14 → MB 6 5x5 SE → 96x14x14 → MB 6 5x5 SE → 96x14x14 → MB 6 5x5 SE → 192x7x7 → MB 6 5x5 SE → 192x7x7 → MB 6 5x5 SE → 192x7x7 → MB 6 5x5 SE → 192x7x7 → Conv 1x1 → 320x7x7 → Pooling+FC

**(g) Ours-L(604M)**

3x224x224 → Stem → 16x112x112 → DS 4 3x3 SE → 16x112x112 → MB 6 5x5 SE → 24x56x56 → MB 6 5x5 SE → 24x56x56 → MB 4 5x5 SE → 24x56x56 → MB 6 5x5 SE → 24x56x56 → MB 6 5x5 SE → 24x56x56 → MB 6 5x5 SE → 40x28x28 → MB 4 5x5 SE → 40x28x28 → MB 6 5x5 SE → 40x28x28 → MB 4 5x5 SE → 40x28x28 → MB 6 5x5 SE → 40x28x28 → MB 6 5x5 SE → 80x14x14 → MB 4 5x5 SE → 80x14x14 → MB 6 5x5 SE → 80x14x14 → MB 4 5x5 SE → 80x14x14 → MB 6 5x5 SE → 80x14x14 → MB 6 5x5 SE → 96x14x14 → MB 6 5x5 SE → 96x14x14 → MB 4 5x5 SE → 96x14x14 → MB 4 5x5 SE → 96x14x14 → MB 6 5x5 SE → 96x14x14 → MB 6 5x5 SE → 96x14x14 → MB 6 5x5 SE → 192x7x7 → MB 6 5x5 SE → 192x7x7 → MB 4 5x5 SE → 192x7x7 → MB 6 5x5 SE → 192x7x7 → MB 6 5x5 SE → 192x7x7 → MB 6 5x5 SE → 192x7x7 → Conv 1x1 → 320x7x7 → Pooling+FC

Figure 1: Discovered architectures (100-600M Flops). "MB $a$ $b\times b$" represents the inverted bottleneck MBConv [1] with the expand rate of $a$ and kernel size of $b$. "DS $a$ $b\times b$" denotes the depthwise separable convolution with the expand rate of $a$ and kernel size of $b$.

Figure 2: Discovered architectures(0-100M Flops). "MB $a$ $b \times b$" represents the inverted bottleneck MBConv [1] with the expand rate of $a$ and kernel size of $b$. "DS $a$ $b \times b$" denotes the depthwise separable convolution with the expand rate of $a$ and kernel size of $b$.

## Appendix C

| Input Shape | Operators | Channels | Repeat | Stride |
|---|---|---|---|---|
| $224^2 \times 3$ | $3 \times 3$ Conv | 16 | 1 | 2 |
| $112^2 \times 16$ | $3 \times 3$ Depthwise Separable Conv | 16 | 1 | 2 |
| $56^2 \times 16$ | MBConv / SkipConnect / ResBlock | 24 | 4 | 2 |
| $28^2 \times 24$ | MBConv / SkipConnect / ResBlock | 40 | 4 | 2 |
| $14^2 \times 40$ | MBConv / SkipConnect / ResBlock | 80 | 4 | 1 |
| $14^2 \times 80$ | MBConv / SkipConnect / ResBlock | 96 | 4 | 2 |
| $7^2 \times 96$ | MBConv / SkipConnect / ResBlock | 192 | 4 | 1 |
| $7^2 \times 192$ | $1 \times 1$ Conv | 320 | 1 | 1 |
| $7^2 \times 320$ | Global Avg. Pooling | 320 | 1 | 1 |
| 320 | $1 \times 1$ Conv | 1,280 | 1 | 1 |
| 1,280 | Fully Connect | 1,000 | 1 | - |

Table 2: The structure of the hypernetwork with additional "ResBlock" operator. The "ResBlock" [2] indicates a residual bottleneck block with kernel size of 3.

| Input Shape | Operators | Channels | Repeat | Stride |
|---|---|---|---|---|
| $224^2 \times 3$ | $3 \times 3$ Conv | 16 | 1 | 2 |
| $112^2 \times 16$ | $3 \times 3$ Depthwise Separable Conv | 16 | 1 | 2 |
| $56^2 \times 16$ | MBConv / SkipConnect / ResBlock / Conv | 24 | 4 | 2 |
| $28^2 \times 24$ | MBConv / SkipConnect / ResBlock / Conv | 40 | 4 | 2 |
| $14^2 \times 40$ | MBConv / SkipConnect / ResBlock / Conv | 80 | 4 | 1 |
| $14^2 \times 80$ | MBConv / SkipConnect / ResBlock / Conv | 96 | 4 | 2 |
| $7^2 \times 96$ | MBConv / SkipConnect / ResBlock / Conv | 192 | 4 | 1 |
| $7^2 \times 192$ | $1 \times 1$ Conv | 320 | 1 | 1 |
| $7^2 \times 320$ | Global Avg. Pooling | 320 | 1 | 1 |
| 320 | $1 \times 1$ Conv | 1,280 | 1 | 1 |
| 1,280 | Fully Connect | 1,000 | 1 | - |

Table 3: The structure of the hypernetwork with additional "ResBlock" and "Normal 2D Conv". The "ResBlock" [2] indicates a residual bottleneck block with kernel size of 3. The "Conv" indicates the standard 2D convolutions with kernel sizes of {1,3,5}.