[Reviews · NeurIPS 2020]

Review 1

Summary and Contributions: This paper is an impressive upgrade of DNA (DNA is the current best-performing method of NAS), which provides an interesting and promising new solution to the timely model ranking problem of NAS. I like this paper. --------------post-rebuttal--------- The authors have answered all of my questions. I am satisfied with the response. As we know, the development of NAS has slowed down in the past months due to the inaccurate architecture rating, which leads to the randomness of NAS. This paper provides an exciting and promising new solution to the timely topic of architecture rating in NAS. I think it deserves acceptance. I thank R3 for the insightful discussions on why knowledge distillation can improve architecture rating. Actually, how to guarantees architecture ranking is an active research topic, which needs more researchers to discuss to re-accelerate the development of NAS. DNA [6], FairNAS, and PCNAS have a conclusion that a smaller search space can lead to a better architecture rating. I also have empirical experiments to support this conclusion. The experiment of R3 in sampling models from --min_flops and --max_flops could also support this conclusion. I also have a theoretical finding, which proves that narrowing the search space could enhance the supernet's generalization ability and results in a sound architecture rating. The block-wisely NAS using knowledge distillation in [6] can modularize the search space into smaller sub-space and improve the architecture rating. FairNAS shows that if the architectures in the search space can be fairly and fully trained, architecture rating could be upgraded. I guess knowledge distillation in this paper can ensure the architectures in the search space to be sufficiently trained, which leads to a good architecture ranking. I appreciate the potential explanation of R3. To sum up, I think this is a good paper. I keep my high rating unchanged.

Strengths: This paper provides an interesting and promising new solution to the timely model ranking problem of NAS.

Weaknesses: As is proved by several papers in ICLR, the problem of architecture rating (ranking) is the main obstacle in NAS, and many of the existing NAS methods is even not better than random architecture selection. The ineffectiveness of NAS is due to the inaccurate architecture ranking. Recognizing this, DNA has solved the architecture problem by using knowledge distillation. This paper improves DNA by removing the third-party teacher. This is a great step. I like this paper very much. I have several minor comments in this paper. First, can the authors provide a theoretical analysis to show that why distilling prioritized paths can lead to accurate architecture rating? Second, what is the validation set used by this paper? As we know, using the validation set to train the matching network will lead to an unfair comparison with the competing methods. Third, DNA does not use AutoAugment. The comparison with DNA is a bit unfair. Please make this detail known in the text.

Correctness: Yes.

Clarity: Yes.

Relation to Prior Work: Yes.

Reproducibility: Yes

Additional Feedback: Please refer to "Weaknesses".


Review 2

Summary and Contributions: The paper proposes a one-shot NAS method aiming to address the problem of insufficient training of the supernet in the traditional one-shot NAS. It proposes to maintain a 'high-performaing network board' to keep the best subnetworks during the search on the fly. For each sampled subnetwork, the method finds a best matched teacher subnetwork from the board through a meta-network, and do the knowledge distillation for the sampled subnetwork using the teacher network. Experiments show that the method achieves superior performance than previous NAS methods.

Strengths: ++ The motivation is sound and the method is intuitive. ++ The teacher network selection from the prioritized paths using a meta network is novel and show great benefits for the search quality. ++ The experiments are extensive. ++ The paper is well-written in general. ++ Codes are provided.

Weaknesses: -- Applying knowledge distillation (KD) to NAS is not novel, even for KD between subnetworks. For example, see [1]. -- Due to the two paths and the meta network, the search time and GPU memory requirement are not advantageous for the proposed method -- Apart from the'random match', it would be good to show the performance of a 'random' baseline in Table 1, i.e. randomly sampled architectures. -- The input to the meta network is the difference between the sampled subnetwork and the subnetworks in the prioritized board. Do authors try different types of the input? -- What is the structure of the meta network? [1] FasterSeg: Searching for Faster Real-time Semantic Segmentation

Correctness: The claims are correct. The method seems correct, supported by extensive experiments.

Clarity: The paper clearly explains its motivation and methodology, although some details are missing, e.g. the structure of the meta network.

Relation to Prior Work: The difference between this work and most of its related works are clearly discussed. But I think reference [1] should be added and discussed as well.

Reproducibility: Yes

Additional Feedback:


Review 3

Summary and Contributions: In this paper, the author proposes a distillation method. The subnetworks within the supernet could teach others while training. The prioritized path is used for boosting the training of subnetworks. SOTA results are achieved.

Strengths: This article is neat and clear writing. The results are strong. The author introduces a meta-network to select good architectures and uses the logits as the soft supervision, this point is novel. This paper belongs to the NAS area.

Weaknesses: The search space is not the same as the google publications but similar to once-for-all. The se-ratio is 0.25 in this paper's code, the expansion rates are {4,6} in this paper and the maximum depth is 5 in every stage, slightly different. Thus, please report #params in Tab. 1. L120. In this paper, the author uses 2K images as the validation set (L212) and use the validation loss to train the meta-network M. I'm curious that the author claim that this step is time-consuming (L159), then how many iterations in total are used for updating M in this paper? The Kendall rank is important, and I prefer more results. BigNAS and once-for-all behave like single-stage NAS, the distillation aims for improving performance. In this paper, the meta-network should solve the ranking problem, and this point should be emphasized. I could not directly find the answer to the question: why KD improves ranking. Firstly, the author uses 30 architectures on ImageNet100, EcoNAS, RegNet might be cited because they use a few models to observe the phenomenon. According to my experience, the randomly sampled 30 models are always with middle-sized, middle FLOPs, which could not cover the global space. But this is enough to demonstrate something. For all the 30 models, we denote the Kendall rank as k@30, I prefer k@10 and k@20. Though the overall Kendall rank is not that high, if the rank on superior architectures is higher, that would be better. The k@10 and k@20 are the rank for the first 10, 20 architectures with the highest accuracies. Maintaining a set of architectures on the Pareto front is similar to CARS. In CARS, the author observed the small model trap, according to Eq. 3, after initialization, only smaller models with higher accuracies will replace the larger models with lower accuracies. Then, where does the large model come from? For example, if the 600M model is not in the initialized set B, where would it come from? According to Eq. 3, the architectures in B will become smaller in every update? L216. If the prioritized path is set to 10, how to select the architectures in Tab. 5 in a single run? Or by setting the desired FLOPs in --flops_minimum and --flops_maximum and search multiple times? Please report the EfficientNet-b0 baseline on the training strategy introduced in 182-186. The EfficientNet-b0 should be 77.3 in Tab. 4 and Tab. 5. This paper uses auto augmentation and ema, the comparisons should be fair. Tab. 7. Please compare with recent sota detectors EfficientDet and Hit-detector. After reading this paper, I want an answer to one question: why distillation helps to rank the architectures? I know that the distillation improves the accuracies for single-stage NAS, for example, [8,9]. [6] indicates different student networks prefer different teacher networks. [7] uses a pre-trained EfficientNet-B7 as supervision because the soft-label produced by B7 may lead to higher accuracy than a one-hot label. They all make sense because the largest subnetwork or pre-trained network are strong enough. In this paper, 10 architecture are used to boost the training of others. The subnetworks may improve their performance under soft-labels, but all the searched architectures are retrained from scratch! How to ensure that the subnetworks with high accuracies under soft-labels (search) could still achieve high performance with the one-hot label (retrain)? Why the supervision from the randomly initialized subnetworks helps ranking? From my point of view, I guess the answer is: 1. Soft-supervision helps training and make subnetworks converge faster than the hard-supervision if we train them for the same iterations, so we need teacher-student training (need figure or related works). 2. Different students prefer different teachers [6], even the largest subnetwork [8] is not the best for different students, so we need lots of teachers (need experiments). 3. Thus, we need to maintain a set of superior architectures (teachers). 4. The superior architectures are used to accelerate the training of different students, so the supernet converges faster (need experiments). 5. The meta-network is used to match them. 6. The ranking would be higher for a better-converged model, as we all know. 7. Thus, the method accelerates the convergence of the whole supernet which indirectly improves ranking. This is my guess, but I can not find the whole logic chain in this paper. The search space is different so I can not say the accuracy is gained from a better search algorithm rather than a better search space. [1] EfficientDet: Scalable and Efficient Object Detection [2] EcoNAS: Finding Proxies for Economical Neural Architecture Search [3] CARS: Continuous Evolution for Efficient Neural Architecture Search [4] RegNet: Designing Network Design Spaces [5] Hit-Detector: Hierarchical Trinity Architecture Search for Object Detection [6] Search to Distill: Pearls are Everywhere but not the Eyes [7] Blockwisely Supervised NAS with Knowledge Distillation [8] BigNAS: Scaling Up Neural Architecture Search with Big Single-Stage Models [9] Once for All- Train One Network and Specialize it for Efficient Deployment After rebuttal: I have read comments from all the reviewers and the feedback. There are still some concerns. 1. [minor] Search space. The search space is different. EfficientNet use 0.04-0.05 se ratio, not 0.25 in rebuttal. The comparison considers FLOPs and acc w/o #params. Nearly all the NAS papers have their own space. Not a big deal. 2. [major] Selection. Actually, there is a small model trap. If not, why use the min_flops parameter? Because the author knows the smaller model has lower accuracy and the smaller model may replace the larger model according to the update equation in the paper. 3. [major] Efficiency. The method is somewhat inefficient. It needs 12 GPU days to search for one model. Other methods, like SPOS or OFA mentioned in this paper search one time for numerous models. 4. [minor] Efficient-b0 in Tab. 4,5. According to https://github.com/tensorflow/tpu/tree/master/models/official/efficientnet, the baseline using AA is 77.1. I'm not sure why the baseline is lower, as all the tricks are used. I have reimplemted and I could achieve 76.8 w/o AA. 5. [major] Novelty. This is not the first paper that introduces KD to NAS. I sill don't know why KD helps ranking. I can only guess that KD helps convergence. This paper uses KD in searching and does not use KD in retraining. Another thing: the models in the board are between --min_flops and --max_flops. I have examined, if the SPOS is trained by sampling models from --min_flops and --max_flops, the ranking is better than sampling models from the global space. So, sampling from a subspace also helps to rank the models in the subspace. This may be another factor that improves ktau except for the KD. Overall, the author addressed some of my concerns. I will raise my score to make a consensus.

Correctness: The concerns are detailed in weaknesses.

Clarity: The paper is well written and easy to follow.

Relation to Prior Work: This paper clearly discussed with previous works.

Reproducibility: Yes

Additional Feedback: Please refer to correctness and weakness.


Review 4

Summary and Contributions: This paper considers the problem of architecture search for computer vision problems under certain constraints, e.g. FLOPs, specifically, on image classification. Unlike the gradient-based architecture search, e.g. DARTS, this paper considers the one-shot neural architecture search, where the idea is to sample and train one subnetwork (a single path within the hypernetwork) at a time. Technically, the authors consider two ideas to improve the training efficiency and performance, e,g. distillation and training a meta network for picking the teacher network (this is different from existing work that uses pre-trained models as teacher). Distillation: during training, a caching mechanism (path board in the paper) is maintained to store the best subnetwork so far, and this will be further used to distill information to the new subnetwork, potentially accelerating the training process. The path board will be updated on-the-fly by the better subnetworks. Meta network is designed to replace the evolution approach in previous work, the goal is to train a matching network that can assign a teacher (from the path board) for distilling information to the new subnetwork. Update (post-rebuttal & discussion with other reviewers). There was broad consensus that this paper makes a good contribution, so I maintain my positive score.

Strengths: The paper is clearly written, easy to understand. The ablation study has clearly demonstrated the effectiveness of the proposed methods, and superior performance has been shown for ImageNet and COCO detection, when comparing with strong baselines under different Flop constraints, e.g. EfficientNet, MobileNet. Utilizing other subnetworks is more convincing than the concurrent works that use pretrained models as teachers for distillation.

Weaknesses: The novelty is somehow limited. It is more towards the engineering direction rather than a principled idea, it includes several small components, and each performs some role in improving the performance.

Correctness: Yes, the authors have experimentally shown the effectiveness of the proposed methods, but it somehow remains unclear to me why the weights trained to optimize one path can be re-used for other paths, I guess this is a question for all these one-shot architecture search approaches.

Clarity: The paper is well-written, easy to understand.This paper is built on the previous approach (SPOS [1]), where the authors have clearly illustrated that, and shown the proposed approach outperform SPOS both in terms of performance and searching efforts.

Relation to Prior Work: This paper is built on the previous approach (SPOS [1]), where the authors have clearly illustrated that, and shown the proposed approach outperform SPOS both in terms of performance and searching efforts. [1]: Guo et al. “Single path one-shot neural architecture search with uniform sampling.”, arXiv preprint arXiv: 1904.00420, 2019

Reproducibility: Yes

Additional Feedback: the authors have provided the codes for reproducing their experiments, though I didn’t try myself, I assume the codes should work.

[Author Response · NeurIPS 2020]

We thank all reviewers for your valuable comments and positive recognition of sound motivation, interesting solution,
strong results, and clear writing. We respond to the concerns point-by-point as below.
——— **To Reviewer #1** ———
Q1.1: Why distilling prioritized paths improves architecture rating? A1.1: Similar to the discussion in DNA [6],
distilling prioritized paths can boost the training of subnets and alleviate the insufficient training issue in weight-sharing
regime. The more sufficient/full training of subnets leads to a more accurate architecture rating [6](Sec.4.3).
Q1.2: The set used to train the matching network? A1.2: The data to train the meta matching network $\mathcal{M}$ is randomly
sampled from the training set, containing 20k images. For more details, please refer to the provided code (file:
*supernet_function.py*, line: 232). We will revise the manuscript to make this point clearer. Thanks for helpful comments.
Q1.3: AutoAugment (AA). A1.3: For a fair comparison, we retrain DNA-d with AA and it obtains up to 0.3%
improvements w.r.t. top-1 acc. on ImageNet. In the revision, we will clarify that DNA does not utilize AA for retraining.
——— **To Reviewer #2** ———
Q2.1: Discussion with FasterSeg. A2.1: Thanks for reminding and we will add the following discussion into Sec. 5.
There are two fundamental differences. 1) FasterSeg searches the teacher model by architecture parameters, while our
method through meta network, allowing each subnet to select its matched teacher model. 2) FasterSeg is a gradient-based
differentiable method, while ours is a sampling-based one-shot method.
Q2.2: Memory and random search. A2.2: 1) Similar to the previous works [5,6,7], utilizing KD for NAS will increase
GPU memory cost. However, KD is capable of speeding up supernet training while improving architecture ranking. 2)
The 3-time random searches of ~450 Flops model gets 75.1±0.7% top-1 accuracy, being 4.1% inferior to our method.
Q2.3: About meta network. A2.3: 1) We tested three types of inputs to the meta network, i.e., the difference, addition,
and concatenation of feature logits. The corresponding top-1 acc. are 79.2/78.9/79.0, respectively. Considering
complexity-accuracy tradeoff, we choose the first option. 2) To keep simple and efficient, the meta network only
contains one fully-connected layer with 1,000 hidden nodes. We will revise the manuscript accordingly. Thanks!
——— **To Reviewer #3** ———
Q3.1: Search space. A3.1: The search spaces of most current one-shot methods are inconsistent, including the
publications from Google. For example, EfficientNet sets se_ratio=0.25, expansion_rate=6, max_depth={5-22}, while
MobileNetV3 are {0,0.25}, {3,4,6} and 4, respectively. Our search space is closely similar to the ones in DNA [6] and
OFA [5], the only difference is expansion_rate: ours is {4,6}, while DNA is {3,6} and OFA is {3,4,6}. Our method
surpasses DNA-c and DNA-d by 1.4 and 1.6 points w.r.t. top-1 acc. on ImageNet. Our method performs comparably to
the finetuned OFA (sometimes slightly better, e.g., 200-300M Flops), but 4.4× faster than OFA, i.e., 12 v.s. 53 GPU days.
The ablation studies in Tab. 1 (the same search space) verify the efficacy of our method. #Params are provided as below.

Q3.2: Iterations for updating $\mathcal{M}$ and Kendall rank $\tau$. A3.2: 1) Updating $\mathcal{M}$ need to calculate

| # | 1 | 2 | 3 | 4 | 5 | 6 |
|---|---|---|---|---|---|---|
| Params | 7.0 | 6.5 | 6.9 | 6.8 | 7.1 | 7.2 |

second-order gradient, which is 7.8× slower than first-order gradient (*i.e.*, 0.5 v.s. 3.9 s/iter).
The total iterations are 1.2M/1K/200*120≃720. 2) We provide the additional results of $\tau@10$

| | @30 | @20 | @10 |
|---|---|---|---|
| $\tau$ | 0.370 | 0.381 | 0.389 |

and $\tau@20$ in the right table. We can observe that $\tau@10$ is slightly superior.
Q3.3: Architecture selection. A3.3: The architectures in Tab. 5 are searched by multiple

| Method | mAP | FLops |
|---|---|---|
| EfficientDet-D0 | 34.6 | 390 |
| Hit-Detector | 36.9 | 1,839 |
| Ours-S | 33.2 | 285 |
| Ours-M | 36.8 | 470 |

runs. In each search, users are required to input the desired *min* and *max* Flops (see Alg. 1).
The prioritized paths in $\mathbb{B}$ are forced to satisfy the Flops constraint, i.e., *Flops*∈[*min*, *max*].
Evolution algorithms, such as NSGA-III in CARS, are not adopted in our method, thus there
is no small model trap issue. Eq. (3) follows Occam's razor principle during model selection.
Q3.4: EfficientNet-B0/Det and Hit-Detector. A3.4: 1) EfficientNet adopted AutoAugment and EMA in the original
paper (Sec. 5.2) and released code (Line 30). For a fair comparison, we retrain EfficientNet-B0 using our setting and
obtain 76.2 top-1 acc., which is comparable to 76.3 reported in ICML19 paper [9] while inferior to 77.3 reported in the
latest ArXiv version. 2) The comparisons with EfficientDet-D0 and Hi-Detector are reported in the upper-right table
and will be updated into revision. 3) The reason why KD improves ranking is discussed in *Q1.1*. Your guess provides a
logical and insightful understanding of this work. We will revise the manuscript in light of these helpful comments.
——— **To Reviewer #4** ———
Q4.1: Novelty. A4.1: This work is a new upgrade of the single-path one-shot method, pushing architecture performance
into new frontiers. The key novelty lies in two aspects: 1) We introduce prioritized paths to remove the the third-party
teacher models from supernet training. The matching between prioritized paths and subnets is meta-learned; 2) Different
from previous methods using evolution search [5,8] to find promising architectures, our method can directly select the
best performing ones from the prioritized paths, which is much more efficient (30× faster than SPOS [8] and OFA [5]).
Q4.2: Why the weights trained to optimize one path can be reused for others? A4.2: Weight sharing across paths are
originally proposed to amortize training cost [13,14]. Feature reuse is the predominant reason for efficient learning
in one-shot NAS. Though now feature reuse lacks clear theoretical guarantee [15,38], it allows ranking architectures,
which would be sufficient if the estimated performance correlates strongly with the actual performance.
Q4.3: About broader impact. A4.3: Our work does not have immediate societal impact, since the algorithm is only
designed for image classification, but it can indirectly impact society. As an example, our work may inspire the creation
of new algorithms and applications with direct societal implications. We will add more discussions. Thanks!

[Meta-Review · NeurIPS 2020]

Reviewers like the proposed method and the experimental results. There are still a number of issues raised by reviewer 3 that could be addressed or discussed to improve the paper.